# Recent Advances in Biogenic Silver Nanoparticles for Their Biomedical Applications

Muskan Goel [1], Anurag Sharma [2] and Bechan Sharma [3],*

[1]  Amity School of Applied Sciences, Amity University, Gurugram 122413, Haryana, India
[2]  Amity School of Biotechnology, Amity University, Gurugram 122413, Haryana, India
[3]  Department of Biochemistry, University of Allahabad, Allahabad 211002, Uttar Pradesh, India
*   Correspondence: sharmabi@yahoo.com

**Abstract:** Owing to the unique property of large surface area/volume of nanoparticles, scientific developments have revolutionized the fields of nanotechnology. Nanoparticles can be synthesized through physical, chemical, and biological routes, where biologically synthesized nanoparticles are also referred to as biogenic-synthesized nanoparticles or bionanoparticles. Bionanoparticles exploit the inherent reducing property of biological entities to develop cost-effective, non-toxic, time-efficient, sustainable, and stable nanosized particles. There is a wide array of biomedical focus on metallic nanoparticles, especially silver nanoparticles, due to their distinctive physiochemical properties making them a suitable therapeutic molecule carrier. This article aims to provide a broad insight into the various classes of living organisms that can be exploited for the development of silver nanoparticles, and elaboratively review the interdisciplinary biomedical applications of biogenically synthesized silver nanoparticles in health and life sciences domains.

**Keywords:** nanotechnology; bionanoparticles; silver nanoparticles; biological entities; in silico applications

## 1. Introduction

In the past decade, the field of nanoscience and nanotechnology has revolutionized science and technology. The term nanoparticle is defined differently by ISO [1], ASTM [1], and IUPAC [2]. The definitions by different organizations consider a variety of attributes such as particle number, concentration of particles, particle size distribution, aggregation/agglomeration of nanoparticles, etc. [1]. However, an internationally recognized definition of nanoparticle has not been established. Apart from the nomenclature, other challenges in the field of nanotechnology include a lack of (a) validated analytical methods and test protocols, (b) reliable exposure and toxicity data, and (c) accurate analytical techniques to precisely characterize nanoparticle morphology [3]. All these developments and challenges suggest robust research and development in nano-research and the importance of nanotechnology in the ever-changing scientific scenario.

In nanoscience and nanotechnology, there is a focus on metallic nanoparticles, especially silver nanoparticles (AgNPs), for their distinctive catalytic activity [4], electrical and thermal conductivity [5], non-linear optical properties [6], and surface-enhanced Raman scattering properties [7]. Moreover, studies suggest that AgNPs have excellent market value in comparison to other nanoparticles from the consumer's perspective [8]. In medicine and biomedical applications, therapeutic effects largely depend on the pharmacokinetics and pharmacodynamics of AgNPs [9]. Due to their biocompatibility and viability, AgNPs act as suitable therapeutic molecule carriers of anticancer [10,11], antimicrobial [12,13], antioxidant [14], and anti-inflammatory [15,16] agents. Moreover, the intrinsic properties of AgNPs, such as binding affinity for various organic molecules and strong absorption, make them a potential candidate for vaccine development and as drug carriers for specific and selective tissue targeting [17]. Even after extended research, there lies a vast gap in

the investigation of AgNPs for biomedical applications. The major hindrance is that the state of AgNPs depends on the medium in which it interacts [18]. For example, AgNPs undergo agglomeration, aggregation, and dissolution on exposure to biological media and biomolecules due to various factors such as organic matter, pH, ionic strength, etc. [18,19]. Another issue is the unavailability of a systemic pattern of comparative analysis of AgNPs for their effects in currently published papers [20,21]. The current studies are focused on synthesizing AgNPs while minimizing these limitations.

AgNPs can be synthesized through three alternative methods, namely physical, chemical, and biogenic routes. The chemical and biogenic routes of AgNPs are comparable, as reducing and stabilizing agents help convert silver ions to AgNPs and further coat particles to maintain size in the nano-size range and anisotropic shape. However, in biogenic routes, biological species act as reducing and/or stabilizing agents in contrast to specific chemicals used in the chemical route. The role of biological species to reduce and/or stabilize and further coat AgNPs makes them an attractive candidate to investigate some prominent concerns such as agglomeration and aggregation. The biogenic methods of synthesizing AgNPs were initiated nearly two decades ago, and currently, more than 1000 biological species are employed in synthesizing AgNPs. It represents a massive success in the biogenic synthesis of AgNPs. This article is specifically focused on the biological route of AgNPs synthesis, where biological species act as a catalyst by reducing, stabilizing, and/or capping Ag+ ions. Various reviews on silver nanoparticles and their biomedical applications are available [22–26]. However, a single elaborative review on the use of all biological species for silver nanoparticle synthesis is not available. This article aims to review various employed biological species such as microorganisms, plants, viruses, and human cell lines involved in the synthesis of AgNPs. The article also throws insight into the biomedical applications of biogenic AgNPs.

## 2. Biosynthesis of AgNPs

### 2.1. Methods of Biosynthesis: Physical, Chemical, and Biological

The synthesis of AgNPs can follow a top-down (physical route) or bottom-up approach (chemical and biological route), as depicted in Figure 1. The top-down approach does not require reducing or stabilizing agents but follows specific techniques reviewed elsewhere [27]. On the other hand, the bottom-up approach requires reducing or stabilizing agents reviewed elsewhere [27]. The reducing and stabilizing agents in the bottom-up approach may be chemical or biological entities classifying this route into chemical and biological methods of AgNPs synthesis, respectively, as depicted in Figure 1. The chemical route of AgNPs synthesis is commonly a three-component system consisting of a metal precursor, reducing agent, and stabilizing agent, where the initial concentration of the metal precursor, the concentration of the stabilizing agent, the potential of the reducing agent, and the molar ratio of metal precursor to reducing agent determine AgNPs size [28]. For example, strong reducing agents (e.g., borohydride) form small-sized AgNPs in contrast to weak reducing agents (e.g., sodium citrate) that produce large-sized AgNPs [28]. The major shortcoming in chemically synthesized AgNPs is the adsorption of certain chemicals on AgNPs surfaces leading to health hazards and toxicity and obstructing their advancement to biomedical use. In contrast to the chemical route, which generally is a three-component based system, the biological route is a two-component system where biological entities both act as reducing and stabilizing agents to reduce metal precursors and stabilize the formed nanoparticles. The use of biological entities both as reducing and stabilizing agents increases their possibility to improve stability, reduce aggregation, increase the reaction rate, and provide efficient purification of AgNPs. Moreover, the adsorption of organic molecules from biogenic species on AgNPs increases their potential for biomedical activity. For example, coating metallic nanoparticles with phytochemicals is suggested to improve the stability of nanoparticles in the external environment and prevent colloidal aggregation [29,30]. In support of this, Mousavi-Khattat et al. stated that though chemically synthesized AgNPs had higher stability after synthesis, their stability decreased with time in comparison to

biogenic-synthesized AgNPs [31]. They further suggested that the synergistic effect of phytochemicals and their coating on AgNPs improve the antibacterial efficacy of biogenic-synthesized AgNPs [31]. A similar study on the functionalization of organic groups on AgNPs surfaces leading to their better anticancer activity in contrast to chemically synthesized AgNPs has been carried out by Kummara et al. [32]. In contrast, Spagnoletti et al. suggested similar bactericidal activity of chemical and biogenic-mediated AgNPs with lower toxicity by the latter (further discussed in Section 2.2.3) [33]. Sreelekha et al. also carried out a comparative study on chemically and biogenically synthesized AgNPs, suggesting that the water-soluble biomolecules adsorbed on AgNPs surfaces provide higher stability to green-synthesized AgNPs and support their higher antioxidant activity [34]. A similar comparative analysis to differentiate the properties of AgNPs produced by the biogenic route from that of the chemical route has been carried out by Veeragoni et al. (discussed further in Section 2.2.5) [35].

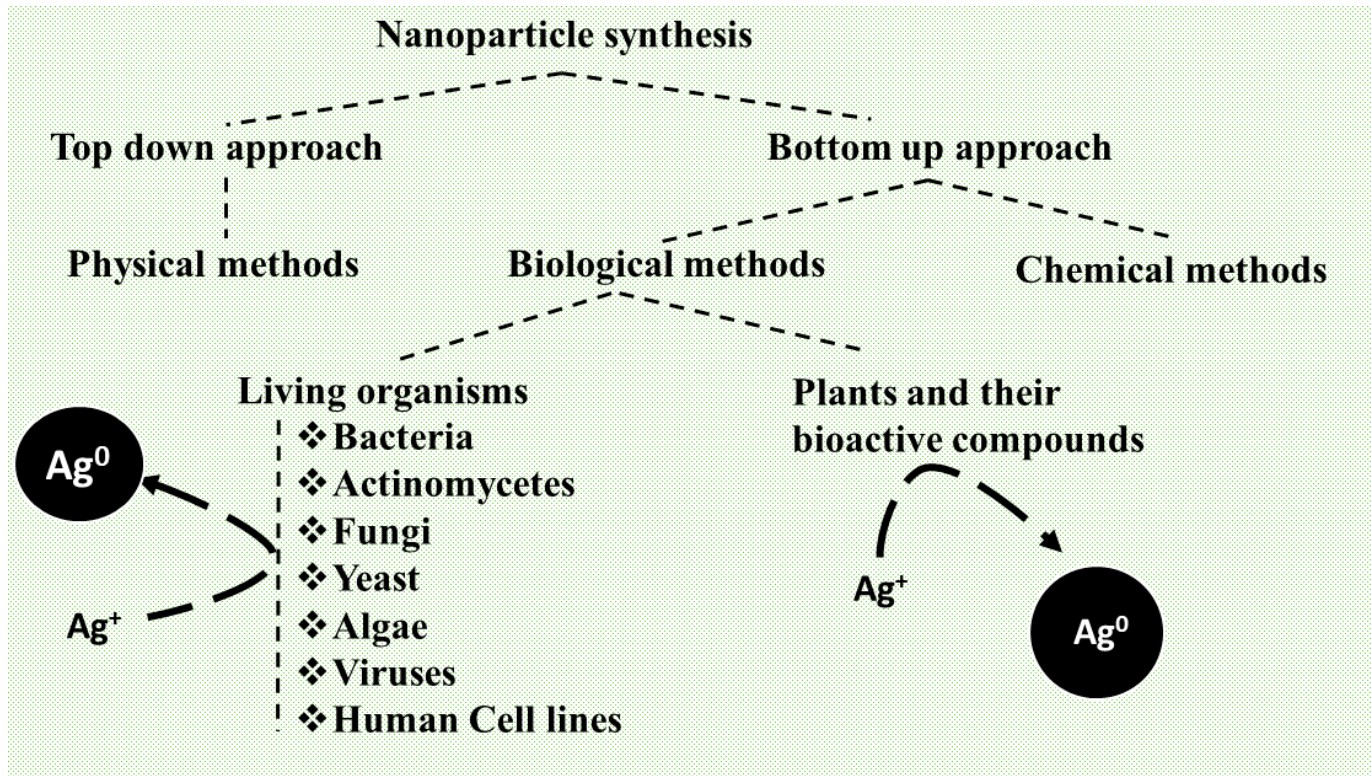

**Figure 1.** The figure depicts various methods of synthesis of nanoparticles with special emphasis on biogenic method of AgNPs synthesis.

The use of biological molecules for synthesizing bionanoparticles is mediated by plants, living organisms, their by-products, or their functional entities. Amongst them, plant-based approaches are substantially preferred due to easy scale-up and higher availability [31]. Furthermore, plants have higher sustainability than other organisms (such as bacteria, actinomycetes, fungi, algae, yeast, viruses, and human cell lines). These organisms require various mandatory conditions (cell line maintenance, cautious handling, may pose a threat of infection to the lab individuals), which decrease their suitability. In this, the reduction of Ag+ to $Ag^0$ is the primary step carried out by the biological entity or its products [36]. Along with the biological entity, the process may also involve ionizing irradiation [37,38], laser irradiation [37], microwave irradiation [39,40], and pulse radiolysis [37]. The biological route is usually energy-efficient; however, employing such techniques may increase energy consumption. Sharing some characteristics of alternative methods, the biological route of AgNPs synthesis can be eco-friendly [23,41], may require less energy [23,42], and support mass production [43], economic feasibility [23,43,44], sustainability, and renewability [45].

The biogenic synthesis of AgNPs can occur intracellularly or extracellularly, as depicted in Figure 2. Due to the diversity and interaction of microorganisms with various metal ions via distinct mechanisms, the exact mechanism of intracellular nanoparticle synthesis is not clear. It is believed that in intracellular synthesis, the positively charged silver ion electrostatically interacts with the negatively charged microbial cell wall or the negatively charged biomolecules (such as enzymes or proteins) present in the microbe's cytoplasm. The microbial enzymes reduce the silver ions, forming small nuclei of distinct morphologies [46–48], and the formed AgNPs can easily diffuse out from the microbial cell wall or cytoplasm. According to the generalized laboratory protocol, the biomass is (a) washed and centrifuged to separate microorganisms from other substances, (b) microorganisms are inoculated with a metal salt solution, and (c) nanoparticles are collected after cell lysis and centrifugation [49]. The extracellular synthesis of nanoparticles depends on the proteins present on the microbe's cell surface or the enzymes secreted by microbes. According to the generalized laboratory protocol, (a) the microorganisms are cultured under suitable conditions for 1–2 days in a rotating shaker, (b) the culture is centrifuged to separate microorganisms from other substances, (c) the microorganisms are inoculated with metal salt solution, and finally (d) the nanoparticles are collected after centrifugation [50].

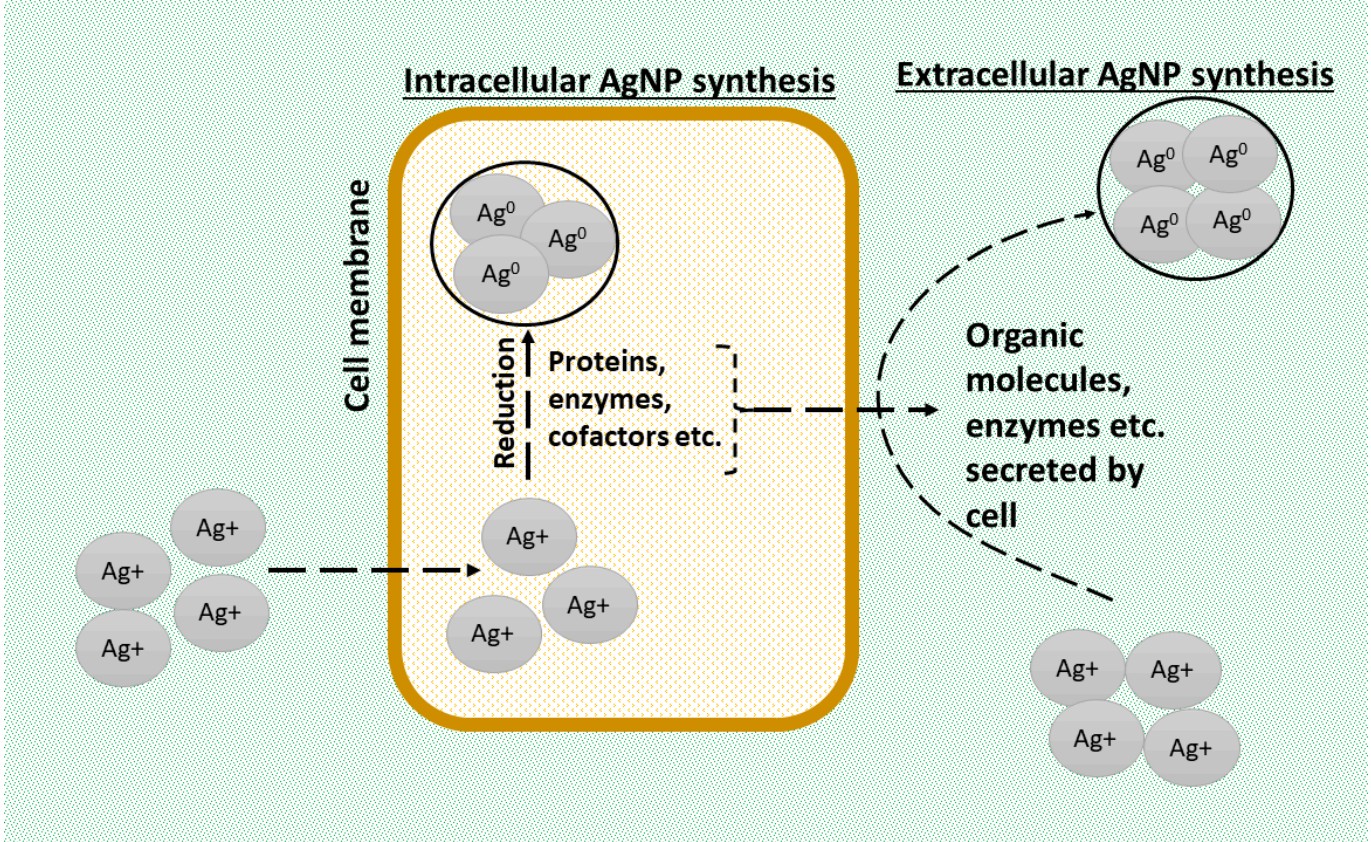

**Figure 2.** The figure depicts intracellular and extracellular biogenic synthesis of AgNPs mediated by biological species.

## 2.2. Biological Species for Nanoparticles Synthesis

### 2.2.1. Bacteria

Bacteria are prokaryotes that have developed a natural defense mechanism to survive continual exposure to toxic metals and environmental conditions [51]. The resistance of bacteria to metals such as silver causes the accumulation of Ag+ in their cell wall, aiding in AgNPs synthesis. Therefore, bacteria's natural mechanisms are exploited for nanoparticle synthesis. The bacterial biomass [52] or supernatant [53] helps in intra- or extracellular AgNPs synthesis. The bacteria's major functional groups involved in synthesis

are carboxylic, hydroxylic, and primary and secondary amides. The major advantages are a high growth rate and easy handling, amongst others. [54,55].

In bacteria-mediated AgNPs synthesis, nanoparticles can be produced from bacterial genomic DNA, culture broth, cell-free supernatants, or protein extracts. Chumpol et al. synthesized AgNPs using both ssDNA and dsDNA and stated that bacterial ssDNA-mediated AgNPs were more stable than dsDNA, as silver ions interacted more efficiently with the nitrogenous bases of ssDNA (produced from denaturing dsDNA) [56]. However, this preparation method involves a three-component system where glucose was additionally required for the conversion of silver ions [56]. Furthermore, their time-dependent synthesis studies suggested that longer time reactions resulted in aggregation, which was confirmed by the shift in SPR peak to a longer wavelength in proportion to time [56]. Saravanan et al. synthesized AgNPs using a culture broth of *Bacillus brevis* (NCIM 2533) and stated that the coating of proteins (from cultural extract) on AgNPs surfaces results in minimal agglomeration [57]. These results are in parallel to studies of Yurtluk et al., who synthesized AgNPs from Bacillus sp. SBT8 and obtained similar SPR peaks [58]. However, Yurtluk et al. also studied the effects of pH and temperature and stated that particle size increases with pH and efficient yield occurs at 33–37 °C [58]. To further analyze the effects of pH, Arzoo et al. isolated 155 strains of Pseudomonas spp. from the rhizosphere, of which three strains, namely SMS13, SMS100, and SMS124, were most efficient in AgNPs synthesis [59]. Arzoo et al. stated that AgNPs (from *Pseudomonas aeruginosa*) were synthesized at the beginning or end of the log phase when most bacterial metabolites are produced and in turn fluctuated the environmental pH [59]. Similar to this, Sable et al. demonstrated the role of nitrate reductase and other enzymes from *Bacillus subtilis spizizenii* in AgNPs synthesis and further suggested that the media components might alter the particle size and optical properties of AgNPs [60]. In contrast to the study conducted by Yurtluk et al., Saleem et al. synthesized AgNPs using bacterial strains (wus1, wus2, and wus5) and suggested that high pH negatively impacts AgNPs synthesis and affects SPR peak [61]. However, the optimum temperature for AgNPs synthesis reported in both studies was parallel to that reported by Singh et al. [62]. Furthermore, Singh et al. suggested the role of water-soluble biomolecules and active enzymes as reducing and capping agents, which were parallel to the results from the above-discussed studies [62]. An interesting study to confirm the role of bacterial proteins was conducted by Li et al., where they used protein extracts of *Deinococcus radiodurans* and suggested that bacterial protein extracts produced monodispersed and spherical AgNPs due to the interactions (reduction and capping) of silver ions with hydroxyl, amine, carboxyl, phosphate, or sulfhydryl groups of proteins of *Deinococcus radiodurans* [63]. The morphological characteristics and applications as studied by bacteria-mediated AgNPs are summarized in Table 1.

**Table 1.** Bacteria-mediated synthesis of AgNPs, their morphological characteristics, and their applications.

| Bacteria | Morphological Characteristics | Application Studied | References |
|---|---|---|---|
| *Escherichia coli strain DH5α* | Spherical, $15.0 \pm 7.6$ nm (TEM) | Antibacterial activity | [56] |
| *Bacillus brevis NCIM 2533* | Spherical, 41–68 nm (SEM) | Antibacterial activity against multidrug-resistant pathogens such as *Salmonella typhi* and *Staphylococcus aureus*. | [57] |
| *Bacillus* sp. *SBT8* | $20.7 \pm 10.5$ nm (SEM) | Antibacterial activity against Gram-positive and Gram-negative pathogens. | [58] |
| *Pseudomonas aeruginosa* | Spherical, 60 nm to 70 nm (SEM) | Antibacterial activity against *Salmonella typhi*, *Shigella dysenteriae*, *Klebsiella pneumoniae*, *P. aeruginosa*, *Proteus mirabilis*, and *Streptococcus epidermidis* | [59] |

**Table 1.** *Cont.*

| Bacteria | Morphological Characteristics | Application Studied | References |
|---|---|---|---|
| *Bacillus subtilis spizizenii* | Quasi-spherical, 10 ± 2 nm (TEM) / Quasi-spherical, 23.8 ± 2 nm (TEM) | Antibacterial activity against Gram-negative (*Escherichia coli*, *Pseudomonas aeruginosa, and Burkholderia cenocepacia*) and Gram-positive (*Staphylococcus aureus*, *Streptococcus faecalis*, and *Clostridium sporogenes*) strains | [60] |
| *E. coli* strain (*wus1, wus2*) *Bacillus* sp. (*wus5*) | Spherical, 10–60 nm (TEM) | Control the nosocomial infections triggered by Methicillin-resistant *Staphylococcus aureus* | [61] |
| *Solibacillus isronensis* | Quasi-spherical, 80–120 nm (TEM) | Antibiofilm activity against *Escherichia coli* and *Pseudomonas aeruginosa* | [62] |
| *Deinococcus radiodurans* | Spherical, 37.13 ± 5.97 nm (TEM) | Anticancer activity against MCF-10A cells | [63] |
| *Streptomyces malachitus* | Round, 19.50 ± 6.72 nm (XRD) | maternal-fetal transplacental transfer assay | [64] |
| *Nisin* | Spherical, 233 nm (TEM) | Evaluate inflammatory activity in macrophage cells | [65] |

### 2.2.2. Actinomycetes

Actinomycetes are Gram-positive, aerobic entities [66] with bacteria-like cell wall composition [67] and fungus-like branched filamentous growth [68,69] found both in soil and aquatic conditions. There is comparatively a limited number of studies synthesizing AgNPs via actinomycetes. Actinomycetes are involved in the intra- and extracellular synthesis of nanoparticles [47]. In AgNPs synthesis, the reduction and consequent formation of nanoparticles occur on the mycelial surface and in the cytoplasm [47,70]. Ag+ ions get trapped on the surface, which then interacts with the functional groups of biomolecules present in mycelia, leading to Ag+ reduction [68]. Streptomyces species, the largest genus of actinobacteria, are involved in the synthesis of AgNPs with numerous biomedical applications. The actinomycetes mediating the synthesis of AgNPs have multiple advantages, such as monodispersity over polydispersity [71], small-sized particles that increase stability and biocompatibility, and biocidal features [72]. Monodispersed nanoparticles have an added advantage, as they suggest better sample-wide uniformity, lower aggregation, and higher stability.

In actinomycetes-mediated AgNPs synthesis, nanoparticles are majorly produced from the culture broth or cell-free supernatant of actinomycetes. Wypij et al. synthesized small-sized AgNPs (that increased stability and biocompatibility) from the *Streptomyces xinghaiensis* OF1 strain, capped with organic compounds such as amino bonds [73]. Similarly, S. et al. synthesized AgNPs from the *Streptomyces hirsutus* strain SNPGA8, where the presence of FTIR-intense bands suggested the role of functional groups, namely alcohols, bromide, iodide, chlorides, and sulfates, in the reduction, stabilization, and capping of AgNPs [74]. However, the above two studies reported the formation of polydispersed AgNPs. To understand the dispersity phenomenon, Mabrouk et al. synthesized AgNPs from *Streptomyces spiralis and Streptomyces rochei* and found that the organism's strain played a critical role in the size homogeneity of nanoparticles [75]. For example, they found that *Streptomyces rochei* could produce monodispersed AgNPs, as a single low-molecular-weight protein was involved in reduction, capping, and stabilization in comparison to the involvement of numerous varied molecular-weight proteins in the production of polydispersed AgNP from *Streptomyces spiralis* [75]. Furthermore, they stated that bactericidal activity is higher for smaller-sized nanoparticles, as they have a higher surface area than large-sized nanoparticles [75]. The morphological characteristics and applications of actinomycetes-mediated AgNPs have been elaborated in Table 2.

**Table 2.** Actinomycetes-mediated synthesis of AgNPs, their morphological characteristics, and their applications.

| Actinomycetes | Morphological Characteristics | Application Studied | References |
|---|---|---|---|
| *Streptomyces xinghaiensis OF1* | Spherical, 5–20 nm (TEM) | Antimicrobial and antibacterial activity | [73] |
| *Streptomyces Hirsutus Strain SNPGA-8* | Spherical, 18.99 nm (TEM) | Antimicrobial and anticancer activity | [74] |
| *Streptomyces spiralis* | Spherical, 20–60 nm (TEM) | Antibacterial activity | [75] |
| *Streptomyces rochei* | Spherical, 5–40 nm (TEM) | | |
| *Streptomyces capillispiralis Ca-1* | Spherical, 23.77–63.14 nm (TEM) | Antimicrobial, antioxidant, and larvicidal activities | [76] |
| *Streptomyces zaomyceticus Oc-5* | Spherical, 11.32–36.72 nm (TEM) | | |
| *Streptomyces pseudogriseolus Acv-11* | Spherical, 11.70–44.73 nm nm (TEM) | | |

2.2.3. Fungi

Fungi are eukaryotic, unicellular, or multicellular heterotrophs that obtain food from dead or living organisms. Fungi have a higher preference than other microorganisms for nanoparticle synthesis [77], as they are easy to grow, handle, and resist agitation amongst other extreme processing conditions. They secrete many proteins, enzymes, and polysaccharides, which play a vital role in synthesizing a diverse range of nanoparticles [78]. The major functional groups involved are carbonyl, amide, hydroxyl, etc. [79,80]. Nanoparticles can be synthesized intracellularly and extracellularly. Intracellular synthesis provides better control over size, while extracellular synthesis is hassle-free due to easy downstream processing steps. Fungi also have an appreciable binding capacity, tolerance, bioaccumulation, and intracellular uptake for silver ions under various experimental conditions [81]. Moreover, the size and structure of fungal-mediated AgNPs can be manipulated by altering pH, temperature, time, and other culture conditions.

Soleimani et al. studied the effect of different pH (5.0, 6.0, 7.0, and 8.0) and temperatures (40 °C and 60 °C) for fungal strains, namely *Beauveria bassiana* (JS1, JS2, and KA75) and *Metarhizium anisopliae*, and stated that 60 °C and pH 7.0 were optimum conditions for the production of small-sized AgNPs in high concentrations [82]. Furthermore, they suggested that isolates KA75 and JS1 produced the most desirable AgNPs [82]. Koli et al. synthesized AgNPs from Monascus red pigments, where sunlight catalyzed the reaction with a reaction time of 5 min [83]. Spagnoletti et al. conducted a comparative study between chemical and biogenic (via *Macrophomina phaseolina*) synthesis of AgNPs and stated that both modes of nanoparticles synthesis had similar bactericidal activity. However, the biogenic-mediated nanoparticles represented lower toxicity in the model organism [33]. Ansari et al. synthesized AgNPs from various fungal species, suggesting the role of carbohydrates (from exopolysaccharides) and not proteins in nanoparticle synthesis [84]. Furthermore, they suggested the highest reduction capacity for *Aspergillus niger* KIBGE-IB36, followed by *Aspergillus terreus* KIBGE-IB35, *Aspergillus flavus* KIBGE-IB34, and *Aspergillus fumigatus* KIBGE-IB33 [84]. Though FTIR analysis was not conducted to confirm the adsorption of functional groups on the nanoparticles surface, a strong SPR peak suggested the formation of stable AgNPs with no aggregation for 3 months [84]. In support of Ansari et al., Li et al. suggested the role of polysaccharides in AgNPs synthesis from *Aspergillus japonicus* PJ01 and stated that reducing sugars helped in reduction, and polysaccharides and proteins supported nanoparticle stabilization [85]. They further studied the role of silver nitrate concentration, pH, and temperature to suggest that the size of AgNPs is proportional to silver nitrate's concentration. Nanoparticle synthesis decreases in extreme alkaline conditions, with the optimum temperature for synthesis being 30 °C [85]. However, the work of Wang et al. was contrary to Li et al., which suggested different optimum pH and temperatures for AgNPs synthesis from *Aspergillus sydowii* [86]. These contrasting results suggested the role of the strain in deciding the optimum conditions of pH and

temperature. The morphological characteristics and applications of fungi-mediated AgNPs are elaborated in Table 3.

**Table 3.** Fungi-mediated synthesis of AgNPs, their morphological characteristics, and their applications.

| Fungi | Morphological Characteristics | Application Studied | References |
|---|---|---|---|
| *Beauveria bassiana* (JS1, JS2, and KA75) and *Metarhizium anisopliae* | Spherical, 23.30, 27.27, 76.61 nm, and 101.34 nm, respectively (SEM) | Antimicrobial and antifungal activity | [82] |
| *Aspergillus niger* | Spherical, 83.36 (DLS) | | |
| *Aspergillus fumigatus* | Spherical, 88.8 (DLS) | Antibacterial activity | [84] |
| *Aspergillus flavus* | Spherical, 208.2 (DLS) | | |
| *Aspergillus terreus* | Spherical, 113.8 (DLS) | | |
| *Monascus pigment* | Spherical, 10–40 nm (TEM) | Antibacterial activity against *Pseudomonas aeruginosa*, *Escherichia coli*, and *Staphylococcus aureus* Antibiofilm activity against antibiotic-resistant *P. aeruginosa* | [83] |
| *Aspergillus japonicus PJ01* | Spherical or irregular, 3.8 nm (TEM) | Antibacterial and antifungal activities | [85] |
| *Macrophomina phaseolina* | Spherical, 40 nm (SEM) | Antibacterial activity, assessment of toxicity in *Caenorhabditis elegans* | [33] |
| *Aspergillus sydowii* | Spherical, 1–24 nm (TEM) | Antifungal activity anticancer activity to HeLa cells and MCF-7 cells | [86] |

### 2.2.4. Yeast

Yeasts are eukaryotic, single-celled organisms, chemoorganotrophs (produce energy from organic matter), widely used in bakery and fermentation processes, that can accumulate a variety of metals. Like fungi, yeasts have a rapid growth process that can be easily manipulated in the laboratory using specific nutrient conditions [87,88]. Yeasts can synthesize nanoparticles intracellularly and extracellularly [89]. The metal ions trapped by yeast undergo oxidation, reduction, sorption, chelation, cell membrane transport, or efflux [87,90]. These processes, by different yeast genera, lead to size and shape-dependent changes in AgNPs. The role of yeast extract as a capping agent helps produce monodispersed nanoparticles that can be easily preserved without precipitation for more than a year [89,91].

Cunha et al. synthesized AgNPs from *Rhodotorula glutinis* and *Rhodotorula mucilaginosa* and suggested that the time required for AgNPs synthesis was proportional to the constituents of the extract that reduced and stabilized the nanoparticles [92]. They further stated that the adsorption of proteins on AgNPs prepared from yeast extract prevented aggregation and sedimentation and enhanced colloidal stability for nearly 15 months [92]. Their study also explained that proteins, ions, and water molecules adsorbed on AgNPs surfaces (in suspended form) caused light scattering, which explained the larger size of nanoparticles when analyzed by DLS in comparison to other techniques [92]. Supporting the work of Cunha et al., Shu et al. synthesized AgNPs from *Saccharomyces cerevisiae* and suggested that biomolecules, namely amino acids, alpha-linolenic acid, and aminobutyric acid, favored controllable size distribution, monodispersity, and stability for nearly a year without precipitation. The morphological characteristics and applications of yeast-mediated AgNPs are presented in Table 4.

**Table 4.** Yeast-mediated synthesis of AgNPs, their morphological characteristics, and their applications.

| Yeast | Morphological Characteristics | Application Studied | References |
|---|---|---|---|
| *Rhodotorula mucilaginosa* UANL-001L | Spherical, 8.89 ± 6.95 nm (TEM) | Antibacterial and antibiofilm properties | [93] |
| *Rhodotorula glutinis* | Spherical, 15.45 ± 7.94 nm (TEM) | Antifungal, catalytic and cytotoxicity activities | [92] |
| *Rhodotorula mucilaginosa* | Spherical, 13.70 ± 8.21 nm (TEM) | | |
| *Saccharomyces cerevisiae* | Spherical, 10.3–18.9 nm (TEM) | Antimicrobial and anticancer activity | [94] |
| Indian Red Yeast Rice | Spherical, 6.81 nm to 30.93 nm (TEM) | Antibacterial and antibiofilm activity | [95] |

2.2.5. Algae

Algae are photosynthetic, unicellular or multicellular eukaryotes, found in water [96] and soil [97]. They can be differentiated into micro- and macroalgae based on their size. Brown algae, green algae, and cyanobacteria are significant varieties of algae that help synthesize nanoparticles [97–99]. Algae are excellent and inexpensive sources of AgNPs production in bulk quantities. The property of algae to develop a specific charge on their surface [97,100] and reduce metals inside and outside the cell makes it a robust biological entity for nanoparticle synthesis. Algae biomass, cell-free extracts, supernatants, and filtrate of broth are used in AgNPs synthesis [99]. The major disadvantages of algae-mediated AgNPs synthesis are the difficulties in the separation of synthesized nanoparticles from the other components involved in the reaction and low production [101,102].

In algae-mediated AgNPs synthesis, the nanoparticles are majorly synthesized from aqueous extract or cell-free supernatant of algae. Rao et al. synthesized AgNPs from fucoidan solution by a microwave irradiation technique and confirmed the adsorption of fucoidan on AgNPs surfaces through a characteristic sulfate group peak, as reflected in the FTIR analysis [103]. They further suggested that the percentage of nanoparticles and fucoidan was 87% and 13%, respectively, and confirmed it through inductively coupled plasma mass spectrometry [103]. In contrast, Bao et al. synthesized AgNPs using *Neochloris oleoabundans* and found the adsorption of no functional groups on the nanoparticles surface, particularly due to the low concentration of cellular materials [104]. However, the reactions under low concentrations suggested the efficiency of reaction by the organism and ease in AgNPs separation. They also stated the relevance of the optimum concentration of AgNO$_3$ (nearly 0.4 mM), optimum pH (between 5 and 7), and extraction time (0.5–10.0 h) for maximum nanoparticle yield [104]. Furthermore, as the reaction was carried out under light conditions, they suggested the dependency on light for the reaction [104]. Husain et al. synthesized AgNPs from Microchaete and suggested that intrinsic capping and stabilization by functional groups of Microchaete prevented the need for further downstream processing [105]. Veeragoni et al. carried out a comparative study analyzing the differences between chemically and *Padina tetrastromatica* mediated AgNPs synthesis, suggesting the role of alcohol, alkane, and nitro groups in chemical synthesis and ketones, aldehydes, and phenol in biogenic method [35]. However, they stated that biogenic-mediated AgNPs were more negatively charged contributing to low aggregation and high stability at different pH conditions and 10% serum biological media [35]. Furthermore, they suggested that the concentration of AgNPs synthesis was pH dependent in biogenic synthesis but not in chemical synthesis, probably due to the impact of ions on bioactive compounds [35]. The morphological characteristics and applications studied by algae-mediated AgNPs are summarized in Table 5.

**Table 5.** Algae-mediated synthesis of AgNPs, their morphological characteristics, and their applications.

| Algae | Morphological Characteristics | Application Studied | References |
|---|---|---|---|
| *Fucus vesiculosus* | Spherical, $36.99 \pm 12.39$ nm (TEM) | Antimicrobial activity | [103] |
| *Neochloris oleoabundans* | Quasi-spherical, 16.63 nm (TEM) | Antibacterial activity | [104] |
| *Ulva flexuosa* | 4.93–6.70 nm (TEM) | Antibacterial activity against two Gram-positive (*Bacillus subtilis*, *Staphylococcus aureus*) and two Gram-negative (*Escherichia coli*, *Pseudomonas aeruginosa*) | [106] |
| *Microchaete* | Spherical, 7 nm (TEM) | Antioxidant, antiproliferative, and apoptotic activities | [105] |
| *Padina tetrastromatica* | spiracle or cubic structure, 166 nm (DLS) | Anticancer activity | [35] |

### 2.2.6. Virus

Viruses are linear, circular, single, or double-stranded nucleic acids with capsid (outermost layer). They hold the capacity to synthesize monodispersed, polyvalent, or symmetric nanoparticles of appreciable surface area and a high aspect ratio [107]. The outer proteinaceous coating of a virus called capsid is majorly involved in binding with metal ions [108]. Some significant viruses employed are brome mosaic virus, cowpea mosaic virus, cowpea chlorotic mottle virus, hibiscus chlorotic ringspot virus, red clover necrotic mosaic virus, tobacco mosaic virus, and turnip yellow mosaic virus [109]. AgNPs can be synthesized with the help of biological substances of virus-like viroid capsules, DNA, multicellular superstructures, and lipid cylinders. Interestingly, different combinations of viral particles can interact with plant extract for the synthesis of bionanoparticles with reduced size in more prominent numbers [110]. Comparatively, virus-mediated synthesis of AgNPs is less common.

### 2.2.7. Plants

Amongst all the biological classes, plants hold the maximum potential for bionanoparticles synthesis, as they are natural sources that can help remove heavy metals from soil and water [111]. This characteristic property of plants is employed for the synthesis of AgNPs. Different chemical components of plants involving various phytochemicals (e.g., catechins, flavones, terpenoids, polyphenols, etc.) support nanoparticles synthesis [112–114]. These phytochemicals are soluble in water and act as reducing and capping agents [115]. The nanoparticles synthesized by plants could be through living plants (intracellular route), plant extracts (extracellular pathway), or phytochemicals (extracellular pathway) [116]. Some methods of plant-mediated AgNPs synthesis are elaborated in Table 6.

In the intracellular route, the plant or its biomass interacts and reduces metal in the aqueous metal salt solution [117,118]. A single plant species can produce polydispersed nanoparticles with variable morphological structures. This diversity is attributed to stabilizing and reducing agents and complex nanoparticle separation and purification procedures [119,120]. In the extracellular route, plant extracts are obtained via hot or cold extraction methods or the Soxhlet apparatus. Phytoconstituents of plants play a vital role in bionanomaterials synthesis. As extracts are first separated and then used for nanoparticle synthesis, the process is called an extracellular method. The process helps in the production of nanoparticles of specific shape and structure, which have a negative potential and are stable in water. Due to variations in phytochemical composition, the process cannot produce monodispersed particles [121].

Phytochemical-mediated synthesis of nanoparticles extends the extracellular route in which specific phytochemical is isolated, quantified, and then used for nanoparticle synthesis. This method helps predict the nanoparticle synthesis mechanism [121]. The process majorly depends on flavonoids and polyphenols and helps control nanoparticle shape and size.

**Table 6.** Plant-mediated synthesis of AgNPs, their morphological characteristics, and their applications.

| Plant | Morphological Characteristics | Application Studied | References |
|---|---|---|---|
| Whole plant | | | |
| *Swertia paniculata* | Spherical, 31–44 nm (TEM) | Antimicrobial activity | [122] |
| *Drosera ittatee Labill var. bakoensis* | Spherical, 21 ± 4 nm (TEM) | Antimicrobial activity | [123] |
| *Brassica oleracea var. botrytis* and *Raphanus sativus* | Spherical, 4–18 nm (TEM) | Antibacterial activity against both Gram-negative (*Escherichia coli*, *Myroides*, *Pseudomonas aeruginosa*) and Gram-positive (*Kocuria and Promicromonospora*) bacteria | [124] |
| *Ajuga bracteosa.* | Spherical, 400 nm (SEM) | Antibacterial, antibiofilm, anticancer activity | [125] |
| *Sida cordifolia* | Spherical, 3–6 nm (TEM) | Antibacterial activity against *Aeromonas hydrophila*, *Pseudomonas fluorescens*, *Flavobacterium branchiophilum*, *Edwardsiella tarda*, and *Yersinia ruckeri*, *Escherichia coli*, *Klebsiella pneumonia*, *Bacillus subtilis*, *Staphylococcus aureus* | [126] |
| Aerial parts | | | |
| *Ephedra procera C. A. Mey.* | Spherical, 20.4 nm (SEM) | Antimicrobial activity against *Escherichia coli* and *Bacillus subtilis* Antioxidant activity Antifungal activity against *A. flavus*, *A. niger*, and *Mucor* spp. Anticancer activity against HepG2 Cells | [127] |
| *Perovskia abrotanoides* | Spherical, 51 nm (SEM) | Antimicrobial activity against *Staphylococcus aureus* and *Bacillus cereus* and Gram-negative bacteria *E. coli* | [128] |
| *Dorema ammoniacum D.* | Spherical, 24.5 nm (TEM) | Antimicrobial activity against Gram-positive (*Bacillus cereus*, *Staphylococcus aureus*) and Gram-negative (*Escherichia coli*, *Salmonella typhimurium*) bacteria | [129] |
| *Lythrum salicaria* | Spherical, 45–65 nm (TEM) | Antimicrobial activity against *E. coli* and *S. aureus* Impregnation of AgNPs into organic nanofibers | [130] |
| *Pistacia terebinthus* (terebinth) | Spherical, 32 nm (SEM) | Antimicrobial, antioxidant, and anticancer effects | [131] |
| *Glaucium corniculatum* (L.) | Spherical, 45 nm (TEM) | Antibacterial activity | [132] |

**Table 6.** *Cont.*

| Plant | Morphological Characteristics | Application Studied | References |
|---|---|---|---|
| *Calotropis procera* | Spherical, 22.14 ± 0.42 nm (TEM) | Antibacterial activity against *Pseudomonas aeruginosa*, *Klebsiella pneumonia*, *Staphylococcus aureus, and Bacillus subtilis* bacteria Antibiofilm and photocatalytic degradation | [133] |
| *Scurrula parasitica* | Spherical, 295, 26.2 ± 0.7 nm (TEM) | Anticancer activity against human lung cancer cells (A549) | [134] |
| *Anthemis atropatana* | Spherical, 38.89 nm (TEM) | Anticancer activity against colon cancer cell lines (HT29) | [135] |
| *Lampranthus coccineus* | Spherical, 10.12–27.89 nm (TEM) | Antiviral activity against HAV-10 virus, HSV-1 virus, and CoxB4 virus | [136] |
| Leaves | | | |
| *Azadirachta indica* | Spherical, 40 nm (TEM) | Antimicrobial activity | [137] |
| *Barleria longiflora* L. | Spherical, 2.4 nm (TEM) | Antimicrobial activity | [138] |
| *Thymus kotschyanus* | Spherical, 22 nm (XRD) | Antimicrobial activity | [139] |
| *Green tea* | Spherical, 11 nm (TEM) | Antimicrobial and antibiofilm activity | [140] |
| *Cyanthillium cinereum* | Spherical, 19.25 ± 0.44 nm | Antimicrobial activity against *Staphylococcus aureus, Klebsiella pneumonia*, biosensor in neurobiology, catalytic properties, antioxidant potential | [141] |
| *Phyla dulcis* | Bead-like, 63–114 nm (DLS) | Antimicrobial activity against *Escherichia coli* O157:H7 (ATCC 43888), *Salmonella Typhimurium* (novobiocin and nalidixic acid-resistant strain), *Listeria monocytogenes* (4b; ATCC 19115), and *Staphylococcus aureus* (ATCC 6538) strains | [142] |
| *Passiflora edulis f. flavicarpa* | Spherical, 3–7 nm (TEM) | Antimicrobial, antioxidant, photocatalytic activity | [143] |
| *Pteris ittate* | Spherical, 17.2 nm (XRD) | Antimicrobial and antivirulence activity against *P. aeruginosa* | [144] |
| Green tea | Quasi-spherical, ~8.3 ± 3.6 nm (TEM) | Antimicrobial and anticancer activity | [145] |
| *Populus ciliata* | Spherical, 4 nm (TEM) | Antimicrobial activity against Gram-positive (*Staphylococcus epidermidis, Streptococcus pyogenes*) and Gram-negative bacteria (*Klebsiella pneumoniae, Serratia marcescens, Pseudomonas pseudoalcaligenes*) | [146] |

**Table 6.** *Cont.*

| Plant | Morphological Characteristics | Application Studied | References |
|---|---|---|---|
| Aloe vera | Spherical, 20.9 nm (XRD) | Antimicrobial activity | [147] |
| Green tea | Spherical, 11 nm | Antimicrobial and antibiofilm activity | [140] |
| Aloe vera | Spherical, 20.9 nm (XRD) | Antimicrobial activity | [147] |
| *Stevia rebaudiana* | Spherical, 50–100 nm (TEM) | Antibacterial activity | [148] |
| *Thymbra spicata* L. (Zahter) | Triangles, hexagons, spheres, and irregular shapes, 70.2 nm (TEM) | Shape-dependent antibacterial and cytotoxic activity | [149] |
| *Cinnamomum tamala* | Spherical, 10 to 12 nm (TEM) | Antibacterial activity against multidrug-resistant bacterial strains (*Escherichia coli* (EC-1), *Klebsiella pneumonia* (KP-1), and *Staphylococcus aureus* (SA-1)). | [150] |
| *Cichorium intybus* L. (chicory) | Spherical, 50 nm (DLS) | Antibacterial activity against Gram-negative (*Escherichia coli*) and Gram-positive (*Staphylococcus aureus*) bacteria | [151] |
| *Barleria buxifolia* | Spherical, 80 nm (DLS) | Antibacterial, antibiofilm, antioxidant, and cytotoxic agent. | [152] |
| *Taxus* | Circular, 15 nm (SEM) | Antibacterial and anticancer activity | [153] |
| *Handroanthus serratifolius* | Spherical, 76.02 ± 3.08 nm (DLS) | Antibacterial activity *E. coli* | [154] |
| *Crescentia cujete* L. | Spherical, 39.74 nm (TEM) | Antibacterial activity against *Bacillus subtilis, Staphylococcus epidermidis, Rhodococcus rhodochrous, Salmonella typhi, Mycobacterium smegmatis, Shigella flexneri*, and *Vibrio cholerae* | [155] |
| *Aesculus hippocastanum* (horse chestnut) | Spherical, 50 ± 5 nm (SEM) | Antibacterial and antioxidant activity | [156] |
| *Litchi chinensis* | Spherical, 5–15 nm (TEM) | Antibacterial and sporicidal activity against *Bacillus subtilis* | [157] |
| Purple heart | Spherical, 98 nm (TEM) | Antibacterial activity | [158] |
| *Taxus* | Circular, 15 nm (SEM) | Antibacterial and anticancer activity | [153] |
| *Datura stramonium* | Spherical, 20.43 nm (DLS) | Antibacterial, antioxidant activity, and DNA cleavage activities | [31] |
| *Lindera strychnifolia* | Spherical, 161, 15.7 ± 1.2 nm (TEM) | Anticancer activity against human lung cancer cells (A549) | [134] |

**Table 6.** *Cont.*

| Plant | Morphological Characteristics | Application Studied | References |
|---|---|---|---|
| *Indigofera tinctoria* | Spherical, 16.46 nm (TEM) | Anticancer activity against lung cancer cell line (A549) Antimicrobial activity against Gram-positive (*Bacillus pumilus, Staphylococcus aureus*), Gram-negative (*Pseudomonas sp, Escherichia coli*) Antifungal activity against *Aspergillus fumigatus*, and *Aspergillus niger* Antioxidant activity | [159] |
| *Cratoxylum formosum* | Spherical, 8.8 ± 0.3 nm (TEM) | Anticancer activity against human lung cancer cells (A549) | [134] |
| *Phoebe lanceolata* | Spherical, 412, 8.8 ± 0.3 nm (TEM) | Anticancer activity against human lung cancer cells (A549) | [134] |
| *Mentha longifolia* L. | Spherical, 20–100 nm (SEM) | Anticancer activity against HCT116 colon cancer cells and Leishmania | [160] |
| *Rubia cordifolia* L. | Spherical, 20.98 nm (TEM) | Anticancer activity, antifungal activity against aflatoxigenic *Aspergillus flavus*, DNA-binding properties, and DPPH and ABTS free-radical inhibition | [161] |
| *Vernonia amygdalina* | Spherical, 41.555 ± 2.488 nm (TEM) | Anticancer activities on the human breast cancer cell line MCF-7. | [162] |
| *Cinnamomum verum* | Spherical, 10 to 45 nm (TEM) | Treatment of Lung Adenocarcinoma | [163] |
| *Berberis thunbergii* | Spherical, 15 nm (TEM) | Anticancer activity against pancreatic cancer | [164] |
| *Aloe arborescens* | Spherical, 40–50 nm (TEM) | Wound healing activity | [165] |
| *Mentha piperita* | Spherical, 35 nm (TEM) | Effect on acetylcholinesterase (AchE) to predict its neurotoxicity. | [166] |
| Aloe vera | Spherical to oval, 10–50 nm (TEM) | chaperone-like activity in the aggregation inhibition of $\alpha$-chymotrypsinogen A | [167] |
| Stems | | | |
| *Picea abies* | Spherical, 78.48 nm (DLS) | Antibacterial, antifungal, and antimitotic effects | [168] |
| *Cannabis sativa* (industrial hemp) | Triangular, rods and hexagonal-shaped, 20–40 nm (TEM) | Antibiofilm activity | [169] |
| *Ceratostigma minus* | Spherical, 16.4 ± 0.3 nm (TEM) | Anticancer activity against human lung cancer cells (A549) | [134] |

**Table 6.** *Cont.*

| Plant | Morphological Characteristics | Application Studied | References |
|---|---|---|---|
| *Mucuna birdwoodiana* | Spherical, 35.4 ± 5.9 nm (TEM) | Anticancer activity against human lung cancer cells (A549) | [134] |
| Roots | | | |
| *Jurinea dolomiaea* | Spherical, cubic, and triangular 24.58 nm | Antimicrobial activity against *Escherichia coli*, *Pseudomonas aeruginosa* Antioxidant activity | [170] |
| *Saussurea lappa* | Spherical, 20.15 nm (XRD) | Antimicrobial activity | [171] |
| *Beta vulgaris* L. | Round, 20–50 nm (TEM) | Anticancer activity | [172] |
| Shikonin | Spherical, 20 nm (TEM) | Anticancer activity in human lung carcinoma cell line A549 cells | [173] |
| *Myrsine africana* | Spherical, 11.4 ± 0.1 nm (TEM) | Anticancer activity against human lung cancer cells (A549) | [134] |
| Tubers | | | |
| Turmeric powder | Spherical, 18 ± 0.5 nm (TEM) | Antimicrobial activity | [174] |
| *Zingiber zerumbet* (L.) | Spherical, 0.2–1 μm (TEM) | Antipneumonial potential in mycoplasmal pneumonia in experimental rats. | [175] |
| *Zingiber officinale* | Spherical, 12 nm | Antifungal activity against *Candida albicans* | [176] |
| *Pueraria tuberosa* | Spherical, 162.72 ± 5.02 nm (DLS) | Anticancer and antioxidant activities | [177] |
| *Alpinia officinarum* | Spherical, 100 nm (TEM) | Effect against the cisplatin-induced nephrotoxicity | [178] |
| *Curcuma longa* | Spherical, 44.9 ± 2.2 nm (TEM) | Study human pterygium-derived keratinocytes | [179] |
| Flowers | | | |
| *Malva sylvestris* | Spherical and hexagonal, 20–40 nm (TEM) | Antimicrobial activity against *Escherichia coli, Staphylococcus aureus, Streptococcus pyogenes* | [180] |
| *Wedelia urticifolia* (Blume) *DC.* | Spherical, <30 nm (TEM) | Antimicrobial activity | [181] |
| *Abelmoschus esculentus* (L.) | Spherical, 16.19 nm (TEM) | Antibacterial and anticancer activity | [182] |
| *Madhuca longifolia* | Spherical, oval, 30–50 nm (TEM) | Antibacterial activity | [183] |
| Fruits | | | |
| *Solanum viarum* | Spherical, oval 2–40 nm (TEM) | Antimicrobial activity against *Bacillus subtilis, Escherichia coli, Pseudomonas aeruginosa, Staphylococcus aureus susp. Aureus, Aspergillus niger*, and *Candida albicans* | [184] |

**Table 6.** *Cont.*

| Plant | Morphological Characteristics | Application Studied | References |
|---|---|---|---|
| Walnut | Spherical, 31.4 nm (DLS) | Antimicrobial, antioxidant, anticancer activity against the MCF-7 tumor cell line | [185] |
| Royal Jelly extract | Spherical, 30–100 nm (DLS) | Antibacterial activity | [186] |
| *Brassica oleracea* (curly kale) | Spherical | Antibacterial, antidiabetic, antioxidant, and anticancer activity | [187] |
| *Benincasa hispida* | Spherical, 27 ± 1 nm (TEM) | Antibacterial activity Antibiofilm activity Anticancer activities against the lung cancer cell line (A549) | [188] |
| Orange | Spherical and ovoid morphology | Antibacterial activity | [189] |
| Pomelo | 35 to 40 nm (XRD) | Antibacterial activity | [190] |
| *Cocos nucifera* (coconut) shell | Spherical, 14.2–22.96 nm (TEM) | Antibacterial activity against *Staphylococcus aureus*, *Listeria monocytogenes*, *Escherichia coli*, and *Salmonella typhimurium* | [191] |
| Elm | Spherical, triangular, rod-shaped, 22.5–30.0 nm (TEM) | Antibacterial, anticancer, and catalytic activity | [192] |
| Grapes | Round-shaped, non-agglomerated 10–40 nm (TEM) | Antibacterial and antifungal activity against Gram-positive (*Bacillus subtilis*), Gram-negative (*Escherichia coli*), and *Candida albicans* wound pathogens. Photocatalytic | [193] |
| Pistachio | Spherical, polygonal 80–100 nm (TEM) | Antibiofilm activity against *S. aureus*, *P. aeruginosa* | [194] |
| *Cornus sanguinea* L. | Spherical, 18 nm (TEM) | Antioxidant and anti-inflammatory activities | [195] |
| *Prunus serrulata* | Spherical, 66 nm (DLS) | Anti-inflammatory | [196] |
| Red onion | Spherical, 12.5 nm (TEM) | antioxidant activity | [197] |
| *Crataegus pentagyna* | Spherical, 25–45 nm (TEM) | Photocatalytic degradation of organic pollutants and in the development of antibacterial materials. | [198] |
| Seeds | | | |
| *Artocarpus hirsutus* | Spherical, 25–40 nm (SEM) | Antibacterial activity against *Enterobacter aerogenes* | [199] |
| *Cassia tora* | 60.78 nm (SEM) | Antibacterial activity | [200] |
| *Catharanthus roseus* | Spherical, 2–15 nm (TEM) | Antibacterial activity against *Escherichia coli* | [201] |

2.2.8. Human Cell Line

Human cells are heterotrophic in nature and require an external source of energy for survival. Recent research conducted on certain human cell lines suggests that they can be utilized in nanoparticle synthesis. The epithelial cells of a healthy individual were used for the synthesis of nanoparticles, in vivo and in situ, without the use of external chemicals as reducing agents. Similarly, cancerous and non-cancerous cells such as HeLa (Homo

sapiens, human), SiHa, and human embryonic kidney-293 cell lines serve as a source of AgNPs synthesis. Human cell-line-mediated AgNPs synthesis is simple, effective, and inexpensive, but less commonly explored [202,203].

### 2.3. Characterization Techniques

Characterization of a material is the analysis of its structure, composition, and physical-chemical properties by determining properties such as size, shape, structure, and surface area. Microscopic and spectroscopic methods can help in nanomaterial characterization, as depicted in Figure 3. Spectroscopic methods include X-ray diffraction (XRD), ultraviolet spectroscopy (UV–Vis), and Fourier transform infrared spectroscopy (FTIR), while microscopic techniques include scanning electron microscopy (SEM) and transmission electron microscopy (TEM). However, these techniques are always used in combination for proper assessment and confirmation of results. The choice of characterization techniques employed depends on the applications specific to the prepared nanoparticles. In comparative studies of biogenic and chemically synthesized AgNPs, the deviation or variability of peaks and intensity has been observed [31,32,34]. The deviation or variability is specifically observed in UV–Vis, FTIR, and XRD results and is attributed to the functionalization of the nanoparticle surface. An extensive review of these techniques can be referred to elsewhere [204].

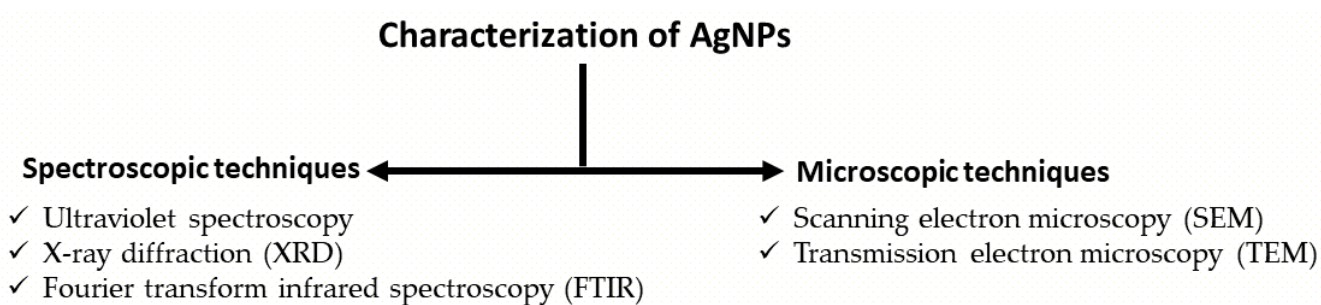

**Figure 3.** Different techniques used for characterization of nanoparticles.

UV–Vis is an absorption-based spectroscopic technique dependent on the Beer–Lambert law. Apart from being fast, easy, sensitive, and selective for different types of nanoparticles, the technique does not require calibration to characterize the colloidal suspension of nanoparticles [205]. The unique optical properties of AgNPs cause them to interact with a specific wavelength. Therefore, AgNPs produce a strong absorption band in UV–Vis spectroscopy. In AgNPs, electrons can easily move, as the conduction and valence bands lie very close to each other. When incident light falls on the surface of AgNPs, it causes the conduction of electrons, resulting in the production of a UV–visible spectrum (also known as surface plasmon absorption band) [206–208]. The UV–visible spectrum or surface plasmon absorption band depends on the shape, size, dielectric medium, and chemical surroundings of AgNPs [206–208].

FTIR is an infrared spectroscopic technique that helps determine a sample's organic and inorganic components and the chemical bonds in the sample. The significant advantages of the technique are its time efficiency, sensitivity, and applicability in all wavelength ranges. However, it is an expensive technique that cannot detect atoms or monatomic ions. The average absorption peaks of AgNPs via FTIR have been observed at 1525 cm$^{-1}$ (N−H), 1670 cm$^{-1}$ (C−O str.), 3070 cm$^{-1}$ (C−H str.), and 3590 cm$^{-1}$ (O−H str.) [209].

XRD is an analytical spectroscopic technique majorly used for crystalline and molecular material characterization, particle size determination, and identification of various compounds [210]. In XRD, a diffraction pattern is formed based on the physical and chemical properties of the material. This diffraction pattern also suggests the sample's purity. XRD peaks have been observed at (111), (200), (220), and (311) (for high concentrations of AgNPs) and (122) and (231) (for low concentrations of AgNPs) [211,212]. The major advantages of the technique are (a) it is inexpensive, and (b) it helps in crystalline structure determination. However, it has size limitations and poor sensitivity.

SEM is a microscopic technique used to determine a material's morphological, topological, and image characteristics [213]. According to the principle, an emitted beam of electrons scatters on the sample's surface and interacts with the sample's atoms. The interaction produces signals (backscattered electrons, auxiliary electrons, and cathodoluminescence) that provide information about the structure and composition. The significant advantages of the technique are high-resolution photo-like images and bulk material characterization. However, it is time-consuming and requires the samples to have electrical conductivity.

TEM is a microscopic technique used to determine the sample's elements and structures. According to the principle, a monochromatic electron beam emits electrons that interact with the sample's atoms. The interaction produces an image visible through CCD cameras or fluorescent screens [214]. The significant advantages of the technique are the production of high-resolution, magnified images with excellent quality and details. However, it is an expensive technique with time-consuming sample preparation (the sample needs to be very thin) [204,215].

## 3. Biomedical Applications

### 3.1. Antimicrobial Activity and Associated Applications

Antibiotic resistance occurs due to mutations in target microorganisms, efflux pumps, and biofilm formation [216], which is a significant problem [217–219], causing the emergence of multidrug-resistant pathogens. The hypothesis that drug-bound AgNPs act as carriers for antibiotics and disrupt bacterial cell walls enabling antibiotic entry is explored to overcome antibiotic resistance. AgNPs are found effective against Gram-positive and Gram-negative bacteria [220], as they work by (a) damaging the cell membrane and its components and (b) inducing cell ROS production that affects the DNA, RNA, and proteins of a cell, as depicted in Figure 4. However, AgNPs have different efficacy due to the presence of thick peptidoglycan in Gram-positive bacteria [221]. Biogenic-mediated AgNPs with antibacterial activity and antifungal activity are discussed in Section 2.

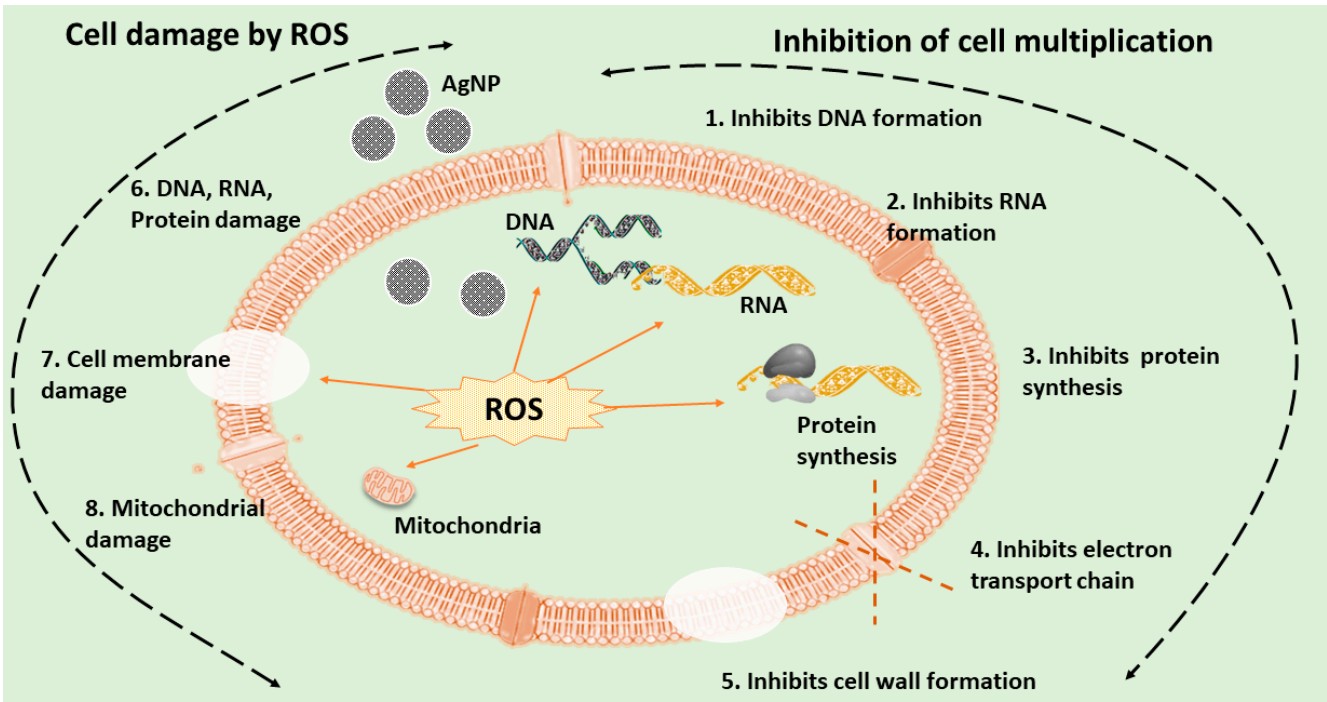

**Figure 4.** Various hypothetical mechanisms followed by AgNPs to inhibit bacterial cell multiplication and cause bacterial cell damage by ROS. Mechanisms 1–5 are followed to inhibit cell multiplication, and Mechanisms 6–8 cause cell damage.

Biofilms represent the attachment and immobilization of a community of microorganisms on a particular surface [222,223] and are an adaptation of microorganisms to survive harsh climatic conditions [224]. This is a causative factor of drug resistance, opportunistic infections, and transmission of infection via dental [225] and biomedical devices [226]. Biogenic AgNPs are found effective against biofilm formation where devices can be coated with nanoparticles. The mechanism of action of AgNPs as an antibiofilm agent is like the antibacterial action depicted in Figure 4. A few effective AgNPs-based antibiofilm agents are discussed in Section 2. One prominent example is the use of AgNPs to coat polyurethane catheters. This helps to decrease the degree of biofilm contamination caused by bacteria.

Moreover, in surgery, especially hip or knee prostheses, there are high chances of bacterial contamination, increasing mortality. This could be avoided by incorporating AgNPs in the prostheses [227–230]. Similarly, incorporating AgNPs in dental instruments and orthodontic adhesives can help prevent bacterial and fungal colonization [228–232].

AgNPs when employed as a vehicle for wound dressing or as drugs for wound healing [233] support repair by exhibiting antibacterial activity, causing immunomodulation and promoting epithelial layer formation and collagen fiber production, as depicted in Figure 5. Various AgNPs with positive wound-healing effects are discussed in Section 2.

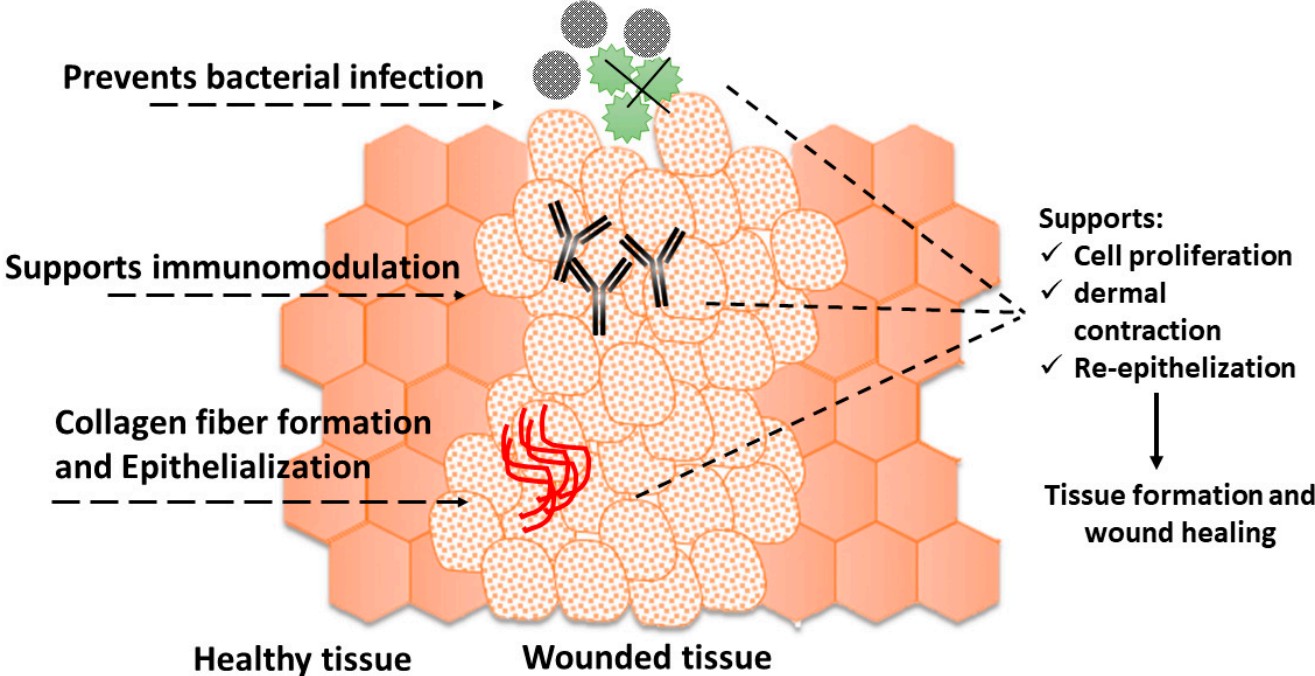

**Figure 5.** The figure depicts various hypothetical mechanisms, namely antibacterial activity, immunomodulation, epithelial layer formation promotion, and collagen fiber production. These mechanisms indicate AgNPs-mediated wound healing.

### 3.2. Antiviral Agents

Viruses pose a significant challenge for life sciences with their remarkable adaptability to the host [234], causing life-threatening diseases. AgNPs act as potential antiviral agents and carriers of antiviral therapies [235] by interacting with viral surface components and blocking viral entry. AgNPs are also believed to prevent viral replication and change the host cell pH, making the environment unfavorable for viruses. Various AgNPs-mediated antiviral mechanisms are depicted in Figure 6. The biosynthesis of AgNPs and their associated antiviral activity are discussed in Section 2.

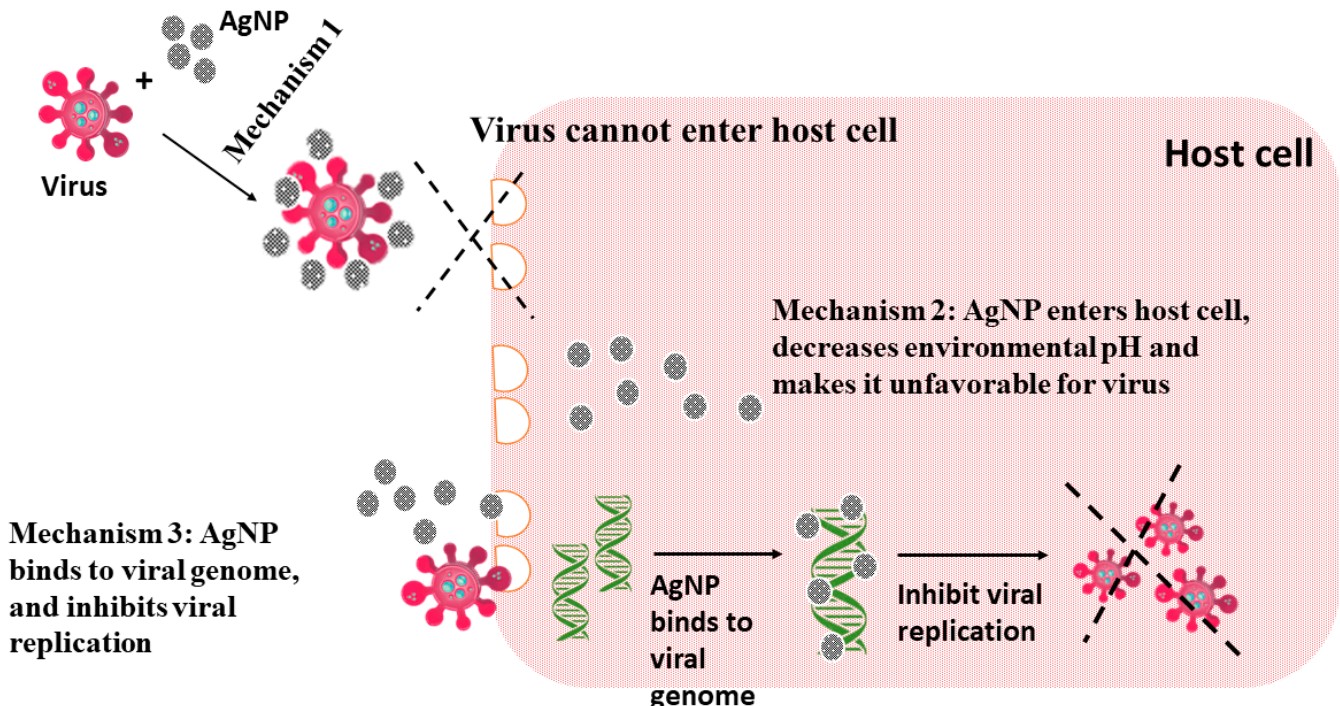

**Figure 6.** The figure depicts various hypothetical mechanisms, namely inhibition of viral entry, decrease in environmental pH in host cell, and inhibition of viral replication in host cell. These mechanisms indicate antiviral activity of AgNPs.

### 3.3. Other Biological Applications of AgNPs

Biosensors, analytical devices to detect an analyte or to measure physiological signals [236], are vital tools in improving therapeutic and diagnostic efficacy. AgNPs have been developed to improve the sensitivity and efficacy of biosensors and are employed in magnetic resonance imaging (MRI), computed tomography (CT) imaging, and photothermal therapy (PTT) [237], as discussed in Section 2.

AgNPs have been shown to act as antidiabetic agents, which act by inhibiting the activities of certain enzymes, such as $\alpha$-amylase and $\alpha$-glucosidase (vital for carbohydrate metabolism) [238]. Various biogenic AgNPs with antidiabetic activity are discussed in Section 2.

AgNPs have also been shown to act as effective anti-inflammatory agents that suppress vascular endothelial growth factor (VEGF), hypoxia-inducible factor-1$\alpha$ (HIF-1$\alpha$), cytokine production (IL-12, TNF$\alpha$), and COX-2 expression [239], as depicted in Figure 7. VEGF is an inflammatory agent that enhances antigen sensitization, T-helper mediated inflammatory cytokines such as IL-4, IL-5, IL-9, and IL-13 [240,241], and HIF-1$\alpha$-mediated bacterial cytotoxicity; and release of proinflammatory factors such as IL-1$\alpha$, IL-6, and TNF-$\alpha$ [242,243]. Thus, the inhibition of these inflammatory factors by AgNPs helps in their anti-inflammatory activity. The biosynthesis of AgNPs and their associated anti-inflammatory activity are discussed in Section 2.

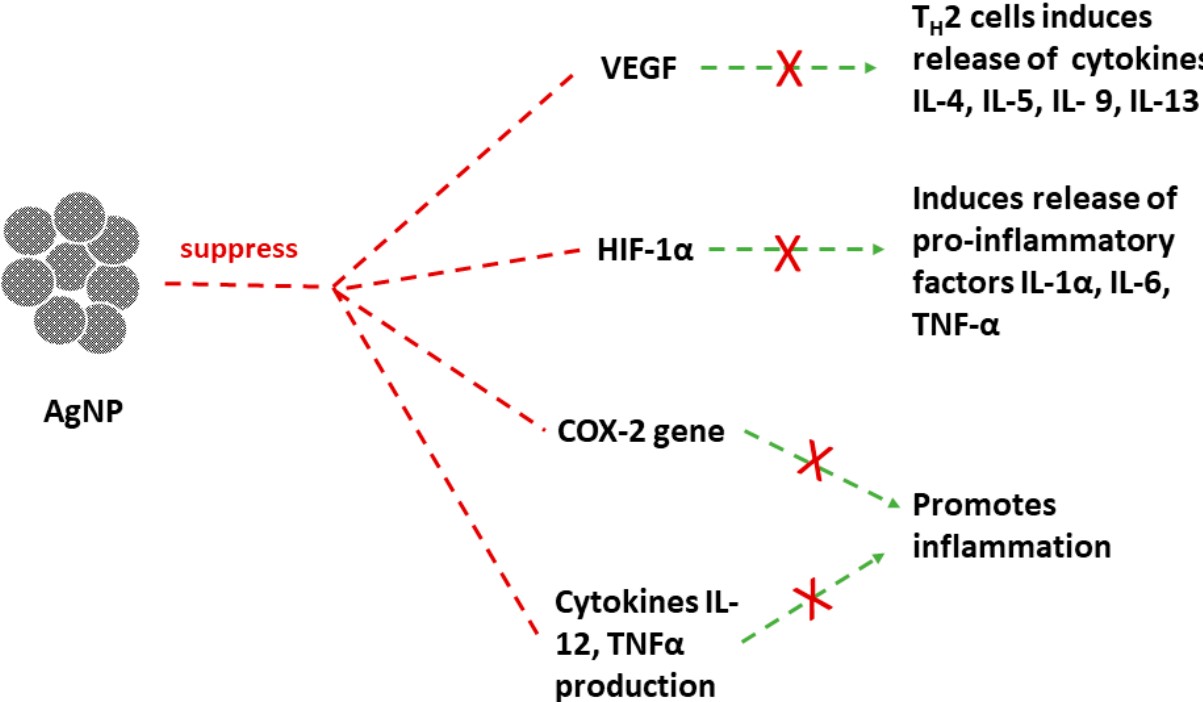

**Figure 7.** The figure depicts AgNPs mediated suppression of inflammatory agents such as vascular endothelial growth factor (VEGF), hypoxia-inducible factor-1α (HIF-1α), cytokine production (IL-12, TNFα), and COX-2 expression. These mechanisms indicate AgNPs -mediated anti-inflammatory activity.

AgNPs act as antioxidants [244–246] as they have the potential to mimic the production of compounds that generate free radicals [247,248]. Apart from free radical scavenging activity, AgNPs also play a role in the absorption and delivery of antioxidant agents [249,250]. The biosynthesis of AgNPs and their associated antioxidant activity are discussed in Section 2.

AgNPs with potential anticancer activity have been employed as diagnostics, therapeutics, theranostics, and drug delivery systems for cancer treatment [251]. For example, in substitution of free doxorubicin, Ag-based nanoparticles in conjugation with doxorubicin are used to inhibit B16F10 cell growth [252,253]. AgNPs biosynthesis and its associated anticancer activity are discussed in Section 2.

Blood coagulation is linked to autoimmune disorders, allergies, injuries, cancer [254], organ failure [255], and thrombosis-associated diseases such as acute coronary syndrome, deep venous thrombosis, pulmonary embolism, stroke, and acute myocardial infarction (AMI). AgNPs are explored for their fibrinolytic activities and antiplatelet aggregation properties. The biosynthesis of AgNPs and their effective anticoagulant activity are discussed in Section 2.

## 4. Toxicity Associated with AgNPs

AgNPs are highly exploited commercial products for their biomedical advantages [256]. The increasing applications of AgNPs have set an alarming concern about their uptake and toxicity. Previous studies have suggested that AgNPs exhibited higher oral or inhalation exposure as compared to their uptake by the skin. One such study has indicated that the AgNPs accumulated in various organs [257]. Research on various animal models has suggested differential accumulation of AgNPs in different organs, i.e., higher AgNPs accumulation in females than in males [258,259]. Nanomaterial toxicity depends on the physicochemical properties and local barriers in the organs. For example, small-sized silver nanoparticles have higher toxicity than their large-sized counterparts [260]. AgNPs toxicity studies have suggested that it leads to pathological changes in various organs, causing damage to certain organs such as the kidney [261,262] and the spleen [262–264].

The results from earlier studies have suggested that AgNPs are linked to inflammation of the blood–brain barrier [265,266] and disruption of the synaptic machinery of neurons [267], thereby affecting neurodevelopment and causing neurotoxicity [268]. These studies further contradict the anti-inflammatory activity of AgNPs. Similarly, the antimicrobial activity of AgNPs also has toxic effects on human cells [269]. In contrast to AgNPs applications for wound healing, it inhibits keratinocyte proliferation [270], and also leads to dermal cytotoxicity. AgNPs are reported to cause blood diseases due to their direct interaction with red blood cells contradicting AgNPs use as an anticoagulant [271,272]. Though AgNPs are studied as anticancer agents, there are numerous reported studies regarding their cytotoxicity related to colon cancer [273] and lung cancer [274,275]. Some elaborated reviews on the toxicity of AgNPs, their effects at the cellular level, their mechanism of cellular effects, and their physicochemical properties leading to their toxicity are out of the scope of this article and are reviewed elsewhere [276,277]. The reported toxicological studies of AgNPs are comparatively fewer than their applications. This, in turn, implies further extensive research on the associated nanotoxicity. Proper models to study AgNPs toxicity with high-throughput analysis and efficient techniques may help in critical evaluation to reach a conclusive remark of the safe and efficient applications of AgNPs. Above all, relevant measures and precautions should be opted to minimize AgNPs toxicity.

## 5. Outlook

Several methods have been reported for the synthesis of silver nanoparticles (AgNPs), using available chemical agents or biological species, investigating the scope and role of the different organic substances in the synthetic process. The knowledge acquired from these interdisciplinary studies has helped to overcome some problems, such as poor stability, aggregation, and agglomeration of the synthesized AgNPs. The primary difference between chemically synthesized and biogenic production of AgNPs is that the first methodology deals with the functionalization of nanoparticles surfaces with organic molecules, causing deviations in the characteristic properties of AgNPs and possibly hindering a biomedical application, whereas the biogenic methods produce biocompatible materials. On the other hand, this functionalization helps to improve the stability and reduce the aggregation, overcoming some important limitations of AgNPs preparation methodologies. Therefore, the advancement in the synthesis methodologies of AgNPs is held back in various ways, preventing its translation to biomedical applications.

The advancement in different methods of AgNPs synthesis is held back in various ways preventing its translation to biomedical applications. Foremost, there are various methods of synthesis, purification, characterization, and validation of data for AgNPs without an established systemic pattern to compare the characteristic properties of AgNPs. It is believed that if research follows a standard protocol such as (a) characterization with all the concerned techniques, (b) finding AgNPs size and morphology by each technique, and (c) comparing results of each technique, etc., it may provide more meaningful analysis. Secondly, AgNPs are merely prepared and studied for their applications with less focus on investigating the primary phytochemicals or organic groups responsible for their applications. Higher attention to studying the role of the organic group that causes deviations during characterization and improves efficacy for their therapeutic applications may minimize the gap in the investigation of AgNPs for biomedical applications. Thirdly, there are numerous AgNPs toxicology studies in contrast to the investigated biomedical applications that require further attention before reaching a conclusive remark on the prospective therapeutic usage of AgNPs. Lastly, the computational advances have proved to be very helpful in the assessment of AgNPs properties, as they help to predict, understand, and validate the data related to biomedical applications. However, compared to the number of reported studies of biogenically synthesized AgNPs, bare minimum articles are available that employ in silico docking and molecular dynamics simulation (MDS) techniques, requiring attention. Altogether, AgNPs may prove to be highly promising in the management of health and diseases and may contribute significantly to the advancement of life sciences research.

**Author Contributions:** Conceptualization, M.G. and B.S.; writing—original draft preparation, M.G.; writing—review and editing, M.G., A.S. and B.S. All authors have read and agreed to the published version of the manuscript.

**Funding:** This research received no external funding.

**Institutional Review Board Statement:** Not applicable.

**Informed Consent Statement:** Not applicable.

**Data Availability Statement:** Not applicable.

**Acknowledgments:** M.G. and A.S. extend gratitude to Amity University, Haryana and B.S. expresses gratitude to the University of Allahabad.

**Conflicts of Interest:** The authors declare no conflict of interest.

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
