# Peer review of "Recent Advances in Biogenic Silver Nanoparticles for Their Biomedical Applications"

_2673-4079, doi:10.3390/suschem4010007_

Round 1

Reviewer 1 Report

The manuscript reviews the green synthesis of silver nanoparticles and potential applications. It follows a plethora of earlier reviews on this rather active field in current nanoscience. Most of recent reviews more specifically focus on a distinct direction and appear thus more profound than the present review. The latter seems like a course of glossary entries as manifested in a large number of extensive tables. The review has a number of shortcomings that require careful revision prior to potential publication.

General aspects:

i) The authors should explain in that sense their review deviates from the flood of others. What makes this review unique? At present, there is no information on other relevant reviews nor an explanation why another review is necessary.

ii) A review should not only be seeing things through rose-colored spectacles. A critical discussion and evaluation is absolutely mandatory. Aspects of nanotoxicity should be taken into account when discussing potentials and perspectives in the medical field.

iii) Extended tables: The data should be condensed. Given size ranges and applications are rather similar in many cases. A better aggregation could substantially help the manuscript to look like a list of glossary entries.

iv) Most paragraphs are organized as follows. At first a new term is defined and briefly introduced by a few general statements. At the end, few remarks on potential links to silver nanoparticles are added. This applies in particular to the parts on characterization and applications. A more extensive discussion, again including critical aspects, would be helpful.

Minor aspects:

i) Page 2: The term “biological template” is misleading. The biological species are used as reducing agent of natural origin but there is no template effect that would make the particles assume a certain shape.

ii) Page 5: The authors say: “Interestingly, bacteria can be used to synthesize various inorganic nanoparticles that are yet impossible to be synthesized by chemical routes.” Without further explanation, providing specific examples, or at least adding references, such a statement is of no value.

iii) Page 6 & 7: “The actinomycetes mediates synthesis of AgNP has multiple advantages – mono and poly-dispersity, stability, biocidal features, etc.” This statement is again rather general. Please explain particular benefits that are to be expected, e.g. from having polydisperse nanoparticles.

iv) Page 11: “ … water that acts as reducing and capping agents.” This expression should be rephrased.

v) Page 22: It is rather surprising that the authors do not consider plasmonic properties when reporting on UV-Vis spectroscopy. Are they aware of the fact that positions of absorption peaks of plasmonic nanoparticles strongly depend on size and shape when they claim that “the average absorption peak of AgNP via UV-Vis is observed at 430 nm.” ?

vi) Page 23: The given wave numbers are specific for ligands but not for AgNP.

vii) It should say “radicals” instead of “radicles”.

Author Response

General aspects:

i) The other relevant reviews available are now referred with reference no. 10-16, 22-26. The importance of this review is addressed on Page 2 (lines 61-64).

ii) Nanotoxicity and its perspective to biomedical applications are now addressed in Section 5.

iii) We understand some tables are extensive, especially plants involved in synthesis of AgNP. We’ve have deleted some of the plant species which do not have specifically reported biomedical applications.

In tables, some of the species are investigated for one activity, others for two three or five.  Moreover, there different combinations of biomedical applications. If we try to concise the tables based on common biomedical applications, it may cause repetition of same species or references or removal of certain biomedical applications specific to species. Similar issue will occur on aggregating species based on size. It may not truly depict the characteristic features of each and every specie.

Moreover, through the tables, the authors are also trying to make evident that plant species are more explored as compared to other biological species. It thereby suggests that research could more be focussed on other biological entities.

iv) The critical aspects on characterization and applications are now addressed in Section 2.3 and Section 5.

Minor aspects:

i) The term biological template is addressed and changed to biological species in the manuscript.

ii) It is now addressed with references and examples.

iii) The issue is now addressed.

iv) The issue is now addressed.

v) The issue is now addressed.

vi) I believe it is regarding the UV-Vis which is now addressed.

vii) The issue is now addressed.

Reviewer 2 Report

The paper presented by Goel et al. it is an interesting well-organized review with an updated bibliography, but there is a need to improve the iconographic quality of the illustrations (especially fig.4), to make the tables more legible and to carefully check the legends of the figures.

Author Response

It is now addressed and Figure 4 has been improvised.

Reviewer 3 Report

Comments to the authors:

In the manuscript authors provide a broad insight into the various classes of living organisms that can be exploited for the development of silver nanoparticles, and elaborately review the inter-disciplinary biomedical applications (investigated through in vitro, in-vivo and in-silico techniques) of biologically synthesized silver nanoparticles in health and life sciences domains. This review has value for the researchers in the related areas. However, the manuscript needs improvement before acceptance for publication. My detailed comments are as follow:

1.      In the introduction section authors should include the relevant articles related to silver nanoparticles properties  

a.       doi.org/10.1002/slct.201900470

b.      doi.org/10.1007/s40089-021-00362-w

If possible other articles also.

2.      If in the characterization part authors should include some images of silver nanoparticles from literature review.

3.      If authors provide some research challenges that will better.

4.      There are few typos errors.

5.      In a small section authors should write a portion why silver nanoparticle not others?

Author Response

1. The aspect is now addressed in lines 36-38 with references 4-7

2. Figure 3 is added depicting different techniques of nanoparticles characterisation.

3. Some research challenges related to nanotechnology (which are also applicable for AgNP) are present in lines 28-32. Challenges related to AgNP are now addressed in lines 47-54.

4. Various typing errors are now addressed.

5. It is mentioned in the introduction on Page 1 and 2 in the lines 35-39 and 41-42. It is now further elaborated in lines 43-45.

Round 2

Reviewer 1 Report

The authors made some important modifications, but could only solve half of my concerns. For example, saying that magnetite nanoparticles cannot be prepared by synthetic routes is absolutely wrong. There is a large number number of reports on that subject.

Overall, the authors could not convincingly explain why another review on this topic is required. For example, I had reviewed rather similar reports in recent months. That's why I cannot recommend publication.

Author Response

With the manuscript, the authors are aiming to provide an updated account of green synthesised AgNP, current progress and major hinderances in a systematic pattern. The primary concern of mere writing of each section and no comparative comments, is a legit point. The authors have improvised by providing contrasting elements in physical, chemical and biogenic methods of preparation; how biogenic methods can help overcome properties of previously prepared NP; deviations in characterisation of biogenic AgNP and reasons etc. The authors realise biogenic method is an alternative method. Henceforth, numerous statements have also been amended. 

We value numerous comments suggested by you which have helped improvise the manuscript at various levels and hope that updated manuscript is relevant to the journal's scope.